# Assessing the Acceptability of Vasectomy as a Family Planning Option: A Qualitative Study with Men in the Kingdom of Eswatini

**DOI:** 10.3390/ijerph16245158

**Published:** 2019-12-17

**Authors:** Philile Shongwe, Busisiwe Ntuli, Sphiwe Madiba

**Affiliations:** Department of Public Health, Sefako Makgatho Health Sciences University, Pretoria 0001, South Africa; philsho@gmail.com (P.S.); busisiwe.ntuli@smu.ac.za (B.N.)

**Keywords:** Eswatini, family planning, vasectomy, acceptance, culture, adoption

## Abstract

The uptake of vasectomy in many countries in sub-Saharan Africa is low. In Eswatini, a kingdom with strong patriarchal norms, the use of vasectomy is at 0.3%. This is despite great efforts to introduce vasectomy and involve men in reproductive health. This study explored the views of men about the acceptability of vasectomy and their willingness to adopt vasectomy as a family planning option. Focus group discussions were conducted with adult men recruited from health facilities located in rural, semi urban, and urban areas in two of the regions of Eswatini. A thematic approach was used to analyze the data. The acceptability of and intention to use vasectomy as a family planning option was very low. Cultural beliefs, societal norms, lack of knowledge about the procedure for vasectomy, and misconceptions influenced the acceptability of vasectomy greatly. The participants could not grasp the concept of a family planning method that is as permanent as vasectomy. However, the decisions to accept or reject vasectomy were influenced by their misconceptions and fears about vasectomy and were not based on facts. To address the need to involve men in reproductive health and improve the acceptability and adoption of vasectomy, planning should be conducted with them and informed by their understanding of their needs.

## 1. Introduction

Vasectomy is a highly effective and safe contraceptive method for couples who want to stop childbearing. It has no side effects and, compared with female sterilization, is a less risky procedure, provides a quicker recovery period, and is incomparable in cost [1,2]. Even though all men who are satisfied with the size of their families are eligible for vasectomy [3], it remains a rejected family planning (FP) option among men. Only 2.4% of men around the world use this method [2], as compared to female sterilization (tubal ligation) which stands at 19% of women worldwide [4]. For instance, vasectomy prevalence is 12% in Northern America, 11% in Oceania and Northern Europe, ranges from 17% to 21% in the United Kingdom and New Zealand, and is 3% in Latin America and the Caribbean regions [5]. Developing countries have a vasectomy prevalence of 2.5%. However, in sub-Saharan Africa (SSA), the uptake of vasectomy remains limited, particularly among African men, with a reported prevalence of less than 1% [2]. This suggests that the uptake of vasectomy among men as a FP option in many countries in SSA is very low, with a prevalence that does not exceed 0.1% [6]. In SSA, Nigeria is the country with the highest reported prevalence at 1%, compared with Namibia (0.8%), Eswatini (0.3%), Botswana (0.4%) and South Africa 0.7% [2].

In SSA, the low uptake of vasectomy is linked to various factors. The literature suggests that women-oriented contraceptives gain more acceptance and uptake than male contraceptive methods due to the notion that FP is a women’s domain [7,8,9]. The most commonly cited factors include myths and misconceptions about the procedure, cultural and religious beliefs, and lack of knowledge or ignorance. The most commonly cited misconceptions include beliefs that vasectomy causes impotence, leads to ejaculatory problems, and is comparable to castration [6]. However, cultural beliefs have taken precedence as a barrier against the uptake of vasectomy [10]. The minimal understanding of the procedure itself as well as fear of complications and fear of losing libido are barriers to the uptake of vasectomy [11]. Furthermore, negative societal perceptions influence its uptake [11,12].

Accurate knowledge and positive attitudes are fundamental to ensuring the informed uptake of vasectomy [13]. This suggests that the lack knowledge about vasectomy and inaccurate information underlie its poor uptake and often influence the way men perceive the procedure [14,15]. Shattuck et al. [2], argue that individuals or couples need accurate knowledge of and positive attitudes toward vasectomy to be motivated to use vasectomy services. However, a systematic review of the literature of program reports on vasectomy found a general lack of awareness about vasectomy and a lack of basic knowledge about the procedure among prospective men and women. The lack of male-friendly services in many setting in SSA leads to most men not knowing about vasectomy let alone having the intention to perform the procedure [2].

Vasectomy is gradually being introduced and is considered as an important development among reproductive health (RH) program designers, policy makers, and population researchers for the overall reproductive well-being of couples globally [16]. However, to improve or increase the uptake of vasectomy as a family planning (FP) method, there is need for enabling policies that will facilitate the inclusion of men not just as default partners of female FP clients but also as equal beneficiaries of FP and RH in their own right [2]. Research has established that engaging men in FP and RH increases couple communication, facilitates male involvement in childcare, and improves relationships [17]. The reality though is that in many SSA settings, few men are involved in issues relating to RH. This perpetuates the traditionalist norm that FP is a women’s thing [7,8,9,18].

Following the Family Planning 2020 (FP2020) movement which began in 2012, to provide women in the poorest countries with access to contraceptives by 2020, some countries in SSA expanded the method mix by increasing the capacity of vasectomy services [19]. In Eswatini, FP is a primary component of the sexual reproductive health (SRH) policy to fast-track efforts to address the sustainable millennium goals to access RH to improve quality of life [20]. In accordance with international recommendations to include men in FP and SRH, in 2015, the Ministry of Health in Eswatini, developed and implemented strategies to involve men in RH by introducing a range of FP methods that are freely available through public and private health facilities. As one of the key strategies to increase the involvement of men in RH, the ministry initiated men’s clinics to increase the uptake of vasectomy. This was crucial in Eswatini, a country with strong patriarchal beliefs [21], where men perceive FP as a women’s thing. Although the country has a National SRH policy, the policy does not refer to the involvement of men in SRH and there is currently no policy on vasectomy as a FP option.

However, efforts to involve men in SRH issues, including on the subject of male contraception, have not been effective. Recent clinic data from health facilities show that the use of vasectomy as a FP method was at 0.0% compared with other FP methods [20]. In 2018, the overall contraceptive use for any method of all women of reproductive age in Eswatini was 51.3% and 65% for married women with a fertility rate of 3.3 births per woman [22]. The FP methods that are widely used are injectables (27.6%), oral pills (13%), male condoms (48.1%), implants (5.1%), and bilateral tubal ligation (4.7%) [22,23]. It should be noted that the introduction of vasectomy in the country was not informed by empirical data, but a need to increase access to FP for women and girls. The need for the service from the perspectives of the health providers and potential users is crucial for the adoption of services. This, however, does not underscore the need for the involvement of men in RH, but highlights the gap in the implementation of strategies. The literature shows that because of a lack of training, health care professionals are more likely to give priority to other services at the cost of a vasectomy because of their attitudes towards and cultural biases against vasectomy [24]. In order to incorporate vasectomy, in the SRH program, there is a need to collect scientific empirical evidence on how men perceive vasectomy. The aim of this study was to explore the views of Eswatini men about the acceptability of vasectomy as FP option in order to achieve a better understanding of the poor uptake of vasectomy in Eswatini.

## 2. Methodology

### 2.1. Study Design and Setting

This paper is extracted from the first author’s dissertation, which was submitted in partial fulfilment of the requirements of a master’s degree in Public Health. This study used focus group discussion (FGDs) to collect data between September and November 2018.

This study was conducted in the kingdom of Eswatini, a landlocked country bordered by Mozambique and South Africa. The distance between Eswatini and South Africa is 430 km. South Africa surrounds Eswatini to the north, west, south and southeast, with Mozambique bordering it on the northeast. Eswatini is divided into four regions: Hhohho, Manzini, Shiselweni, and Lubombo. The population of Eswatini is 1,093,238 people, and, among those, 562,127 are females and 531,111 are males. The country is a patriarchal kingdom, with strong patriarchal norms and clearly defined gender roles for men and women. Men in Eswatini are considered the heads of their families and culturally, women are minors, their minority being reinforced by marriage. In Eswatini, cultural practices take precedence and polygamy and the payment of lobola (bride price) are largely practiced. These traditional laws and cultural practices subordinate women, who are considered not to be equal to their spouses [21].

The participants in this research project were recruited using purposive sampling from health facilities located in two of the regions of Eswatini. The study setting included three health facilities, and the first data collection site was a public facility located in a rural setting in the Hhohho region. The second site was a satellite public clinic in a densely populated semi-urban area, also in the Hhohho region. The third data collection site was a privately owned urban clinic located in the Manzini region. This clinic is reserved exclusively for male clients; the clinic provides medical male circumcision services, tuberculosis (TB) screening and treatment, sexually transmitted infections (STI) treatments, FP for males, curatives, and laboratory services.

The three data collection sites were chosen to satisfy the desire for geographical triangulation. They represent a rural clinic, semi-urban clinic and an urban clinic. According to the Ministry of Health, a significantly larger proportion of health workers in urban and rural facilities are female—this is of particular significance among nurses and midwives, a female predominant profession [25]. However, the men’s clinic is manned by four nurses—two male and two female nurses—one male doctor and male phlebotomists. 

Flyers with health promotion messages on FP methods, on vasectomy as a FP option, as well as a chart on surgical procedure for vasectomy were distributed in the study sites before the schedule date for data collection. These flyers were used in morning health talks that are routinely delivered to clients. On the days of data collection, the first author (PS) delivered the talks. After the talk, brochures on vasectomy were distributed to those that were literate; the brochures were in English and SiSwati languages. The participants were recruited after the morning talk; those who volunteered to participate were informed that they were going to form a focus group.

Participants were recruited if they were aged 35 years or older. Since this study sought to assess the acceptability of vasectomy as a FP option, it was crucial to ensure variation in the information gathered. The sample include men who were single and married, younger and older, and lived in urban, rural, and semi urban areas.

### 2.2. Data Collection

Focus group discussions were preferred because they provide a safe environment for participants to speak about sensitive information such as vasectomy [26]. This is a sensitive issue in Eswatini, where the cultural practices and norms eschew open discussion of sexual matters. The focus group allowed the men to speak in a specific context in their specific culture, as opposed to expressing their individual views. Focus groups can be an empowering process for open discourse if the participants are homogenous in terms of gender or ethnicity, and create a conducive environment, particularly when a complex topic is discussed. The focus groups allowed the men to provide both individual opinions and community perspectives about vasectomy [21], whereas diversity in the group in terms of age and other socio-demographics provided evidence of important group differences and a deeper understanding of the barriers against the acceptability and uptake of vasectomy in the community [27].

The first author, who was the moderator of the focus groups, is trained in qualitative methods as well as in facilitating FGDs. She is an experienced health professional grounded in issues of RH and the cultural beliefs of her communities. These qualities made her acceptable to the men in the discussion of vasectomy, a topic that would generally be taboo for discussion in the presence of a woman. Moreover, these attributes equipped the moderator to facilitate the FGDs in a culturally acceptable manner, using the local language (SiSwati). Since cultural practices and norms view discussions about sexual matters as inappropriate, the moderator played a role in ensuring that the participants felt comfortable by providing a clear explanation of the purpose of the FGDs and assured the participants of the confidentiality of the discussion.

The moderator used a semi-structured focus group guide with open-ended questions to facilitate the FGDs. The open-ended questions addressed broad questions about the topic including (1) their views about FP, (2) the female spousal use of FP, (3) previous knowledge and awareness about vasectomy, (4) views about vasectomy, (5) acceptability of vasectomy, (6) willingness to use vasectomy in the future, and (7) their views of what would facilitate their involvement in SRH. Over and above these broad questions, the moderator used probes and asked follow-up questions. The use of open-ended questions and probes allowed the participants to express their opinions using concepts unique to their language. All FGDs were conducted in a private room to allow the participants to speak freely. They were recorded after obtaining consent from the participants, and each session lasted for about 35 to 45 min. Eight FGDs were conducted across the three sites—four were conducted in the semi-urban site, two in the rural site, and two in the urban men’s clinic. On average, each focus group comprised 7 to 8 men, making a total of 54 men. At the end of each FGD, background information including age, marital status, employment and level of education was collected from the participants, using a short demographic tool.

Ethics approval was obtained from the Ethics Committee of Sefako Makgatho Health Sciences University (MREC/H/21/2018: PG) and the Eswatini Research Ethics Committee. Permission was also obtained from the relevant regions and facility authorities. Participation in this study was voluntary and all the participants signed an informed consent form.

### 2.3. Data Analysis 

All of the authors took part in the data analysis, as guided by a thematic analysis approach [28]. The first author transcribed the audio-recorded FGDs verbatim, translated them into English, and edited and formatted them in preparation for entering them into NVivo 10, a qualitative data analysis computer software package [29]. All authors were conversant with SiSwati and English. They reviewed the data for accuracy by playing back the recordings while repeatedly reading a few transcripts to familiarize themselves with the data. The translated data were open coded in order to identify initial emerging themes, which were grouped and categorized in order to begin developing a codebook. All the transcripts were subsequently imported into NVivo in order to apply codes. The authors engaged in a rigorous process to define and reach consensus on the emerging themes and subthemes, which were used to present the findings.

Various methodologies were engaged to ensure trustworthiness: peer debriefing sessions were held after each FGD, the local language was used to facilitate the sessions, field and interview notes were taken, and the transcripts were transcribed verbatim. Data showing the process, records, and findings were kept as an audit trail [30].

## 3. Results

### 3.1. Description of Study Sample

The sample comprised of 54 adult men with a mean age of 48 years. Most (44%; n = 24) were in the age group 41–50 years. Most were married (80%; n = 43) and the rest were not in a marriage relationship with their sexual partners. Of the participants, 28 reported having one wife, 13 had two wives, and two had three wives. The average number of children amongst the men was five. Regarding their educational status, most (n = 32) had some form of education, 35% (n = 19) had a tertiary education and 24% (n = 13) had completed high school. Almost one-quarter (22%; n = 12) reported having no formal education. A total of 41 were employed, while 13 reported being unemployed (Table 1).

### 3.2. Themes Identified

Thematic analysis of the FGDs yielded eight themes reflecting the participants’ views about the acceptability of vasectomy and these were: family planning is beneficial, family planning is a women’s thing, a lack of knowledge about vasectomy, the perceived health risk, misconceptions about vasectomy, religious beliefs, and a need for focused health services for men.

#### 3.2.1. Family Planning is Beneficial

The focus group interviews revealed that the participants perceived FP as beneficial to the family. They stated that FP enables the family to determine the number of children they desire to be able to provide for the family.

“*My views are or let’s say life and the way I live, suggests that it is better to do family planning. I am saying this because when you look at the life we live these days, there is education and retrenchment, scarcity of employment while children are also at school. All this makes it hard to actually feed and clothe, as there are no opportunities for handy work. It is best to work together with your wife to provide for the as there are no opportunities for handy work. It is best to work together with your wife to provide for the family as it is not the same as in the past*”.(FGD 1 rural clinic, 45 years old)

#### 3.2.2. Family Planning Is a Women’s Thing

Despite the cited benefits of family planning, some of the participants believed that FP is a women’s thing.

“*As men we can see that the whole issue of family planning was meant for women, because even when you enter the facilities you can see that it is meant for women. All available facilities have been rightly arranged for women. Ehh this whole thing when you look at it even the nurse profession was meant for women. I tend not to understand how then this whole issue involves men because eish we have released the women that they can go to the clinics for family planning*”.(FGD 5 semi urban clinic, 49 years old)

#### 3.2.3. Lack of Knowledge about Vasectomy

The discussions revealed that other than the traditional method of withdrawal and the male condom, most of the participants were not aware of male methods available to them. For some, the first time they heard about vasectomy was when the moderator was given an opportunity to talk about it in the morning health talks delivered before rendering services to clients.

“*The condom is better we have seen its advantages instead of vasectomy which we do not even know*”.(FGD 3 semi-rural clinic, 50 years old)

“*I for one I do not know anything about vasectomy; it is my first time hearing of it what we know is the use of condoms*”.(FGD 3 semi-rural clinic, 49 years old)

“*The men in urban areas are the ones who are informed in such issues especially on family planning. We as men do not know that we can do vasectomy, what we know is that it is all on the shoulders of the women*”.(FGD 2 rural clinic, 60 years old)

#### 3.2.4. Cultural Beliefs

Eswatini, the study setting, is a culture-dominant country, where men subscribe to strong cultural practices and beliefs. One of the cultural beliefs is that men are the heads of their families. The participants in the FGDs felt that their roles as men in their families and society would be diminished by their undergoing vasectomy. Cultural and societal expectations such as having many children, especially male children that would carry the family name into the future, were pivotal in how they perceived vasectomy.

“*No mam, it will not be right because the man is the head of the household, if they do the vasectomy it would mean I would have shot myself on the foot because there will never be a child in the house then what can you do*”.(FGD 2 rural clinic, 45 years old)

“*I can understand the conversation is good, but then to the men it gets to be hard in the sense that I may agree with my wife that she wants two children, just like me I am young and have one child and you find that upon agreement the wife passes away when I have sterilized. When the wife passes as we all know that a man cannot live without a partner, I find a new wife and the problem will arise because I have sterilized.*”.(FGD 1 rural clinic, 45 years old)

One other concern for the participants regarding vasectomy is that it is a permanent method of FP. They were uncertain about the future after vasectomy uptake, particularly with regards the lineage of their families. They explained that in the Eswatini culture, men can have many wives and they felt that after vasectomy, taking a new wife would be problematic because they would not be able to reproduce with all their wives. The future need for children influenced their negative attitudes towards vasectomy.

“*The wife can pass on together with the children, what then can you say? Once a man goes through vasectomy what can you do when there is a new wife? There will be a need to have a new wife who will have children in the marriage*”.(FGD 2 rural clinic, 47 years old)

“*Because in our culture men can have more than one wife. When I then find myself in polygamous family after I had sterilized; I have, three or four wives how do I explain myself? It will not be easy*”.(FGD 4 semi urban clinic, 55 years old)

“*On another note if all my children pass away then what will you do if you have done vasectomy?*”. (FGD 8 urban men’s clinic, 46 years)

They explained that in their culture, if a brother dies, one is expected to take care of the brother’s wife to continue with the family lineage. The lineage in the context of the Eswatini men extends to the households of extended families in their clans. They were explicit that they would not undertake vasectomy because they have a responsibility to ensure that the lineage in their brother’s house continues. They have an obligation to take care of their brother’s family when their brother has died, should they be assigned that role.

“*If you do the vasectomy maybe in the long run as years go by they tell you that sir can you go and take care of that household, there is no woman who would agree to have a man who cannot have children*”.(FGD 2 rural clinic, 57 years old)

“*If it comes to my attention that my younger brother cannot have children, according to my culture I should step in and help my brother’s family with children. I cannot look at his wealth being taken by someone who is not from the family. I cannot do that if I have undergone vasectomy*”.(FGD 8 urban men’s clinic, 46 years)

The belief that every family must have a son in order for the family name to continue made vasectomy undesirable to the participants. The pressure to meet the expectations of culture made it difficult for them to support or play an active role in accepting vasectomy.

“*You do appreciate your girls but I do want the boy so that they continue with the family legacy. A girl does not give birth to children of the same surname as themselves. That is why if your wife has five girls you can go back to your wife’s family to ask for another wife who can help the first wife who has not been able to give birth to a boy who will run with the family name*”.(FGD 8 urban men’s clinic, 55 years old)

#### 3.2.5. Perceived Health Risk

The data revealed that the participants had fears of possible health risks attached to vasectomy. They were skeptical that such a delicate procedure would not have negative effects on their sexual health. The fear of losing libido was prevalent amongst the participants and influenced their perceptions of the procedure. They were also skeptical and doubtful of vasectomy, given that it protects against unwanted pregnancies only, and not against the risk of STIs.

“*When I sterilize what will happen to me …, will I get an erection because the sperms are responsible of making an erection so that it has a passage to go out? The filling of the scrotum on its own is a problem when it does not get empty. It can lead to big testes or even cancer of the testes*”.(FGD 6 semi urban clinic, 54 years old)

“*After doing vasectomy will I still have the same urge/erection and be able to engage in sex as before or I will have less strength as I’m now an ox that has been castrated*”.(FGD 7 urban men’s clinic, 45 years)

#### 3.2.6. Misconceptions about Vasectomy

The data revealed that misconceptions about vasectomy resulted in the negative attitudes of the participants towards the procedure. One of their foremost concerns regarding vasectomy was its perceived negative impact on their sex lives. The most common myths include beliefs that vasectomy causes impotence and ejaculatory problems. They equated vasectomy with castration.

“*I was only making an example on the things that happen if a cow has been castrated. I mentioned before that the manhood does not function when a cow is castrated, same goes for men whom their manhood does not erect*”.(FGD 5 semi urban clinic, 38 years old)

“*If you look at sterilized male cow it gains so much weight, how do we as men get away from that without gaining weight easily*”.(FGD 2 rural clinic, 51 years old)

The participants felt that vasectomy would promote promiscuity. They believed that following vasectomy their sexual performance would be affected, which might force their wives to be promiscuous to satisfy their sexual needs. They further believed that vasectomy would encourage indulgence in unprotected sex by men, particularly as the main deterrent to such behavior, “pregnancy”, would no longer be a problem. They were concerned that this would affect their marriage and sexual health in a negative way.

“*I feel like this will encourage a lot of cheating because if you marry your wife while you have undergone vasectomy and you do not want children. In short, I am saying she will go out to find what she needs. What happens there?*”.(FGD 6 semi urban clinic, 39 years old)

“*Looking at things I see a lot of families separating because if a man knows that I am now free because I will not bring home any children they start to have extramarital affairs*”.(FGD 2 rural clinic, 62 years old)

#### 3.2.7. Religious Beliefs

The data also revealed that the participant’s religious beliefs and practices were a barrier to the acceptance and uptake of vasectomy. They believed that vasectomy was a sin and forbidden by religion.

“*According to the Bible it says multiply and be as many as the sand of the sea, how will I do that if as a man I am now sterile; that will be defying the word of God as I am a born-again Christian*”.(FGD 8 urban men’s clinic, 49 years old)

“*I think this is killing; this is one way of killing because if I am no longer having children when God did not stop me from giving birth its actually killing. Is it not a sin as I am now supposed to put on a condom, and every ejaculation falls in it? Will God agree that I have killed a person*”.(FGD 6 semi urban clinic, 53 years old)

#### 3.2.8. Need for Focused Health Services for Men

In response to what they perceived would facilitate their involvement in SRH, they felt that there was a need for focused health services for men such as special clinics for men, with services rendered by male health professionals, and awareness campaigns focused on men’s health. They stated that special clinics for men would increase utilization of services by men.

“*A clinic specifically for men where you are able to talk about all your problems, if in all the clinics or hospitals there were rooms designated for men where on top of coming with different illnesses but also where men are educated on issues concerning them*”.(FGD 7 urban clinic, 50 years old)

“*When you go to the clinic for men like the one situated in xxx, but then you walk in and find a woman sitting there and she asked what you came for. It is another thing that is depressing because if I am suffering with something and find the woman who will say that I should show her my penis then you feel ashamed*”.(FGD 7 urban clinic, 62 years old)

“*The health sector should deploy male nurses to assist male clients so that they are free to talk about male problems, not the very young nurses deployed to the clinics making it hard for male clients*”.(FGD 6 semi urban clinic, 37 years old)

## 4. Discussion

This study explored the views of Eswatini men about vasectomy as a FP option. This study found that, consistent with findings from studies in SSA, the participants lacked basic knowledge and accurate information on vasectomy [2]. Most of the participants had no prior knowledge of vasectomy as a FP method despite the relatively high level of literacy among the men. Of the 54 participants, 19 had a tertiary education and 13 had completed high school. The lack of awareness about vasectomy and lack of basic knowledge about the procedure is consistent with findings from other studies in SSA and developed countries [2]. According to the United Nations [31], vasectomy is the least known modern FP method in most low-resource countries. This research also found what other studies documented, that because of their limited levels of knowledge concerning vasectomy, the participants had a poor grasp of the procedure itself, which greatly influenced their misconceptions about its complications and contributed to its low acceptability [2,32]. Perry et al. [13] argue that accurate knowledge and positive attitudes are fundamental to ensuring the informed uptake of vasectomy.

The lack of knowledge and awareness of vasectomy could be attributed to the poor promotion of vasectomy as a men’s FP option. The literature suggests that health providers often have poor knowledge about vasectomy or have attitudes towards and cultural biases against it, which affect what they say and do when they interact with clients [18,24]. It is imperative that providers are trained in client-oriented service provision for underutilized methods, such as vasectomy, to increase access and knowledge [2].

Intention to use vasectomy among the study participants was very low; other studies in SSA found that a small percentage of men reported even considering a vasectomy [33]. Although almost all of the participants found vasectomy unacceptable, the younger participants (11 out of 54) and those who were unmarried (11 out of 54) were totally opposed to it and could not even comprehend why such a method of FP could be promoted. This suggests that the packaging of messages about vasectomy should take into account the different target groups within communities. Vasectomy should be promoted as an effective and safe contraceptive method for couples who want to stop childbearing. This would prevent the total rejection of the method.

The data showed that one of the biggest concerns for the participants regarding vasectomy was its perceived negative impact on their sex lives. Besides the fear of the loss of their manhood, the men also feared losing respect in their homes, and believed that their poor sexual performance might result in their spouses seeking other men to satisfy their sexual needs. Likewise, the findings of other studies were that men had fears that a lowered libido would cause them to be unable to adequately meet the conjugal rights of their wives [9,10,34].

The poor grasp of the procedure for vasectomy resulted in the participants assuming that vasectomy would bring about certain health risks. For example, they equated vasectomy to castration and believed that the procedure for vasectomy would have the same effect that castration has on cows. Shelton and Jacobson [18] is of the view that myths and misconceptions play a key role in the acceptance of vasectomy as a FP method. They argue that even when men and women know of the method, their knowledge is filled with misconceptions; especially that vasectomy is castration or makes men weak. Consistent with other studies [2,6,32,35] the participants expressed concerns that vasectomy would weaken their sexual ability. Social norms concerning FP in general, require an enabling environment that begins with the creation of comprehensive awareness and the provision of information to increase the knowledge and understanding of the different types of reproductive services that are available.

The kingdom of Eswatini, where this study was conducted, is a patriarchal country with ingrained cultural practices, where men are viewed as the heads of the family and women are considered as minors. It is, therefore, not surprising that cultural beliefs and practices were the main barriers against the acceptability of vasectomy as a method of FP. This is consistent with the findings of studies conducted in other developing parts of the world, which documented how culture presents itself as a barrier against vasectomy [2,35]. Polygamy is commonly practiced in Eswatini, and 15 of the 54 participants in this study were in polygamous marriages—in which, having many children is a norm. The average number of children amongst the participants was five. The permanency of the procedure was the main concern of the participants, who felt that their roles in the family and society would be degraded. Sharma et al. [36] report negative societal perceptions of vasectomy, where men who opt for vasectomy are looked down upon, are perceived as impotent, and are viewed as less than male.

Future uncertainty was identified as another barrier against the acceptability of vasectomy among the participants, who were not certain about their future need for children. The high value placed on children in society made it difficult for them to accept a permanent procedure such as vasectomy. Similar concerns have been reported by previous studies, that men are often concerned about undergoing vasectomy because they are not certain about the future need for children [37,38].

It was found, consistent with other studies conducted in SSA [39,40], that religion and religious beliefs influenced the acceptability of vasectomy. The participants referred to the Bible, claiming that it says they must reproduce and fill the earth. This is consistent with the findings of other studies, which reported that men believed that getting a vasectomy was a sin [41]. Likewise, religious beliefs were the main barrier against decision making in FP, particularly amongst Muslims [12,42].

In many settings in SSA, women were found to have an influence on men’s perceptions of and acceptance of vasectomy. The few participants who found vasectomy acceptable and were willing to undergo vasectomy in the current study, would do so conditionally. They stated that they would undergo vasectomy only if their spouses agreed to the procedure. The majority of the men in other studies also indicated that their acceptance of vasectomy was dependent on the consent of their spouses [2,6]. Other studies that were conducted in SSA settings attest to the fact that spousal disapproval influences the uptake of vasectomy in many societies [36,43]. This suggests that women play an essential role in the vasectomy decision-making sphere, even in a patriarchal society such as Eswatini.

### Limitations

This study is subject to some limitations; one of the key limitation is that it was limited to adult men of 35 years and older including those who are single or without a partner.

Secondly, this study was performed with a sample of men with the exclusion of their partners or women in general; therefore, the results should be taken cautiously and cannot be generalized to women.

There are implications for research—for example, a larger quantitative sample study could determine the factors that are associated with poor acceptability of vasectomy as a FP option. Furthermore, this study could have been expanded to include the female perception of vasectomy and the level of their support should their partners elect to have such a procedure. 

The strength of this study is the inclusion of a diverse group of men to talk about sexual and RH, which is regarded as taboo in the country. The findings do not reflect any inhibitions to openly discuss vasectomy in view of the fact that the moderator was a female. 

## 5. Conclusions

Cultural beliefs and societal norms influenced the acceptability of vasectomy to a greater extent. This is consistent with the belief systems of a country that has strong cultural beliefs and practices. The findings demonstrate uniformity of belief where male vasectomy is concerned. The research found that, regardless of the variations in the level of education, age, and marital status of the participants, there was consistency of thought about vasectomy among all of them, which led to low acceptance.

The findings suggest that the homogeneity of their thoughts around vasectomy was influenced by their cultural beliefs, their lack of knowledge and awareness about the procedure for vasectomy, and misconceptions. They could not grasp the concept of a family planning method that is as permanent as vasectomy, since their culture and social norms expect that marriage should produce children. Furthermore, their view that any form of family planning is women’s responsibility influenced what they thought about family planning and vasectomy in particular. Lastly, their decisions to accept or reject vasectomy were influenced by their misconceptions and fears about vasectomy and were not based on facts.

### Recommendations

There is a need to develop an enabling environment at a policy and programming level to facilitate the integration of vasectomy in FP programming to increase the offering of vasectomy. The role of health professionals in this regard is crucial and calls for their training to understand the importance of constructively involving men in RH. The training of male health professionals to render services in men’s clinics is advocated in settings in SSA, where cultural norms and biases are prevalent.

One other key strategy to address the lack of knowledge and awareness about vasectomy is to develop information, education and communication (IEC) materials with gender-transformative messaging. The health promotion messaging should respond to cultural beliefs and societal norms in an undertaking to involve men in FP issues. This should culminate in focused awareness campaigns for men to educate men and women about vasectomy and other female-only FP options.

The fact that few participants in this study were open to the idea of undergoing vasectomy suggests that addressing the fears and misconceptions that men hold about vasectomy will improve its uptake. However, the lessons learned in this research is that the initiation of the strategy to involve men in SRH did not involve men from the beginning to select the approaches to effectively reach them.

To address the need to involve men in SRH and improve the acceptability and uptake of vasectomy, planning should be conducted with them and informed by their understanding of their needs. In this study, men wanted men’s only clinics staffed by male health professionals.

## Figures and Tables

**Table 1 ijerph-16-05158-t001:** Description of the study sample.

Variable	Number	Percent
**Age**		
35–40 years	11	20
41–50 years	24	44
51–60 years	13	24
61–70 years	6	11
**Marital Status**		
Married	43	80
Single	6	11
Living with partner	3	6
Divorced	2	3
**Level of education**		
Tertiary	19	35
High school	7	13
Secondary	12	22
Primary	13	24
No formal education	3	6
**Employment status**		
Employed	41	76
Unemployed	13	24

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
