# Peer review of "Assessing the Acceptability of Vasectomy as a Family Planning Option: A Qualitative Study with Men in the Kingdom of Eswatini"

_ijerph, 2019, doi:10.3390/ijerph16245158_

Round 1

Reviewer 1 Report

The manuscript, "Culture as a barrier to the acceptability of vasectomy as a family planning option: A qualitative study with 3 men in the Kingdom of Eswatini " does provide an important contribution to the literature on men and vasectomy, in that examination of the men in the Swazi population has been under-studied.  Therefore, there are important merits to consider However, several factors reduce this reviewer's enthusiasm for the manuscript in its current form.  First, there is not a robust description of “what motivates researchers to undertake the study”.  Second, the international appeal is missing from this work. What can be learned from this Swazi sample? Third, describing possible social and cultural consequences from the literature would be better.  The data collection period is unknown, no information about how the sample was chosen, single, unmarried sample also were included in the study. No information about the inclusion and exclusion criterias, no information about the questions used in the focus group. Overall, if this is to be largely a simple thematic analysis with purposive sampling (no adjustment according to age, education level etc.), it should be noted as such. I would encourage the authors to consider ways to further organize the review of literature so that themes are apparent as are the gaps - it would be easier to see how this study adds to what is already known about the topic.

I summarise my review below for your information:

The introduction session needs to be more succinct to highlight the key findings of the existing literature on the topic area. Please explicitly state the research gap, explain “What is the incidence rate of contraceptive use among women and men?”, “the prevalence of other men related contraceptive methods such as condom and withdrawal”, “fertility rate in the country”, “What is the health policy about vasectomy in the country” etc. The data collection method needs to be strengthened. In particular, i) explain how you selected the sample is it purposive sampling or randomly selected for data collection. ii) elaborate the results of the guide for focus group interviews what did you use to collect data, is it a semi structured questionnaire, did you have an expert opinion, if you had an expert opinion what were the advices (what kind of advices were given for guiding the focus interviews). What was suggested by the experts about interview time? It is not clear: I assume the researchers asked potential participant men after their appointments to participate in the study. Please highlight the key findings in the results session. My impression is the reproductive health care services or attitudes of health care professionals towards vasectomy may be causing the low acceptability of vasectomy in the country due to the finding of low knowledge about vasectomy. It would be helpful to clarify “were single men or divorced men p without partner informed about vasectomy in a routine way?, For example: one participant has expressed very important statement “ Even though we are hearing something new, we did not know that there is something that we can 175 do on family planning”. It sounds like he had no contact or take any services related to reproductive health care. Is it common in that region of the country? The Discussion should focus more on discussing the key findings with the existing literature. Cultural and social values were repeated throughout the discussion section. It is important to discuss how other factors in the studying country shaped the findings. There are several places needs citation and

There are several limitations to this study. First, the study was performed with a sample of men including single or without partner. Therefore, the results should be taken cautiously and cannot be generalized to married women. The study limitation needs more elaboration, particularly types of response bias. Please add any recommendation to cope with such limitations in future studies. Side note - I would not end with limitations - give the article a negative ending - I would also add a small section on strengths, then end with a take home message such the need for programming... This article needs a section for recommendations: There is a sentence in the conclusion for recommendation “it is crucial that health professionals 383 establish awareness campaigns focused on men’s family planning options, to educate men about 384 vasectomy and other family planning options. This would begin to address the need to involve men 385 in reproductive health and improve the acceptability and uptake of vasectomy. Furthermore, health 386 promotion strategies should respond to cultural beliefs and societal norms in an undertaking to 387 involve men in FP issues.” However; the recommendation is not helpful and simply very general. Please detail what kind of strategy? What are your other suggestions related to your findings for this specific society? Please highlight the key messages in the conclusion session. It would be very nice if authors elaborate implications to practice, research, or professional development. I also think that a section on implications should be added and concluding comments strengthened that would link this study to the health care professionals. It would benefit from expanding a little on how these findings could be used to operationalize improvements to men's health care in the international area.

Best.

Author Response

Response to reviewer’s comments

We thank the reviewers for their valuable comments, we have responded to all the comments highlighted in this document and in the document. All revisions or additions are highlighted in blue.

Comments and Suggestions for Authors

The manuscript, "Culture as a barrier to the acceptability of vasectomy as a family planning option: A qualitative study with 3 men in the Kingdom of Eswatini " does provide an important contribution to the literature on men and vasectomy, in that examination of the men in the Swazi population has been under-studied.  Therefore, there are important merits to consider 

However, several factors reduce this reviewer's enthusiasm for the manuscript in its current form.  

First, there is not a robust description of “what motivates researchers to undertake the study”.

Second, the international appeal is missing from this work. What can be learned from this Swazi sample? Third, describing possible social and cultural consequences from the literature would be better.  

The data collection period is unknown, no information about how the sample was chosen, single, unmarried sample also were included in the study. No information about the inclusion and no information about exclusion criteria,

We apologize for the lack of clarity; we added text on the inclusion criteria. We recruited the participants if they were aged 35 years or older. Since the study sought to assess acceptability of vasectomy as a FP option, it was crucial to ensure variation in the information gathered. We include men who were, single and married, younger and older who lived in urban, rural, and semi urban areas.

No information about the questions used in the focus group.

We added text to clarify that we used a semi structure focus group guide with open-ended questions to facilitate the FGDs. The open-ended questions addressed broad questions about the topic including; 1) their views about family planning, 2) the female spousal use of family planning, 3) previous knowledge and awareness about vasectomy, 4) views about vasectomy, 4) acceptability of vasectomy, 5) willingness to use vasectomy in the future, and 6) their views on what would facilitate their involvement in SRH. Over and above these broad questions, the moderator used probes and asked follow up questions.

Overall, if this is to be largely a simple thematic analysis with purposive sampling (no adjustment according to age, education level etc.), it should be noted as such.

We appreciate the comment, but the study was conceptualized mainly as a description of the views of the participants on vasectomy using simple thematic analysis. However, in the discussion we refer to the sociodemographic data that we collected. For example, we argue that the younger men were completely against vasectomy as compared to some of the older men who would consider vasectomy. In the conclusion, we argue that all the participants had similar thoughts and belief about vasectomy in spite of the differences in their educational attainment.  

I would encourage the authors to consider ways to further organize the review of literature so that themes are apparent as are the gaps - it would be easier to see how this study adds to what is already known about the topic.

I summarise my review below for your information:

The introduction session needs to be more succinct to highlight the key findings of the existing literature on the topic area. Please explicitly state the research gap, explain “What is the incidence rate of contraceptive use among women and men?”, “the prevalence of other men related contraceptive methods such as condom and withdrawal”, “fertility rate in the country”, “What is the health policy about vasectomy in the country” etc.

We revised the introduction as per the comments. We added two paragraphs and articulates the focus of the introduction and stated the rationale for doing the study.

The country has a National SRH Policy but the policy does not refer to the involvement of men in SRH and there is no mention of vasectomy.

The data collection method needs to be strengthened. In particular, i) explain how you selected the sample is it purposive sampling or randomly selected for data collection.

The participants in this research project were recruited using purposive sampling from three health facilities located in two of the regions of Eswatini. We added text to clarify that of the three study sites, the first was a public facility located in a rural setting, the second was public clinic in semi-urban area, and thee third was a privately owned men’s urban clinic

Elaborate the results of the guide for focus group interviews what did you use to collect data, is it a semi structured questionnaire,

We added text to clarify that we used a semi structure focus group guide with open-ended questions to facilitate the FGDs. The open-ended questions addressed broad questions about the topic including; 1) their views about family planning, 2) the female spousal use of family planning, 3) previous knowledge and awareness about vasectomy, 4) views about vasectomy, 4) acceptability of vasectomy, 5) willingness to use vasectomy in the future, and 6) their views on what would facilitate their involvement in SRH. Over and above these broad questions, the moderator used probes and asked follow up questions.

Did you have an expert opinion, if you had an expert opinion what were the advices (what kind of advices were given for guiding the focus interviews). What was suggested by the experts about interview time? It is not clear:

The focus group guide was developed from literature review and finalised after several role-plays between the first author and the last author.

I assume the researchers asked potential participant men after their appointments to participate in the study.

We provide details on the procedure followed to recruit the men for the focus group discussion. Flyers with health promotion messages on FP methods, on vasectomy as a FP option, as well as a chart on vasectomy surgical procedure were distributed in the study sites before the schedule date for data collection. On the days of data collection, the first author delivered the talks and distributed brochures on vasectomy in English and SiSwati languages. The participants were recruited after the talks, and those who volunteered were included in the FGDS.

Please highlight the key findings in the results session. My impression is the reproductive health care services or attitudes of health care professionals towards vasectomy may be causing the low acceptability of vasectomy in the country due to the finding of low knowledge about vasectomy. It would be helpful to clarify “were single men or divorced men without partner informed about vasectomy in a routine way?, For example: one participant has expressed very important statement “ Even though we are hearing something new, we did not know that there is something that we can  do on family planning”. It sounds like he had no contact or take any services related to reproductive health care. Is it common in that region of the country?

We did not ask the participants about the views of health services but on their thought oh what could improve their involvement in SRH. Their narratives did not indicate that they received information about vasectomy from the health professionals; however, they recommended the provision of men’s only clinics for all their needs. We added a theme on their views.

The Discussion should focus more on discussing the key findings with the existing literature. Cultural and social values were repeated throughout the discussion section. It is important to discuss how other factors in the studying country shaped the findings.

We have revised the discussion and limited the focus on culture based on the comments from both reviewers.

There are several limitations to this study. First, the study was performed with a sample of men including single or without partner. Therefore, the results should be taken cautiously and cannot be generalized to married women. The study limitation needs more elaboration, particularly types of response bias. Please add any recommendation to cope with such limitations in future studies. Side note - I would not end with limitations - give the article a negative ending - I would also add a small section on strengths, then end with a take home message such the need for programming...

Thanks for the comments; we have added text on the implications of the findings and on the strength of the study and placed the limitations before the conclusion.

This article needs a section for recommendations: There is a sentence in the conclusion for recommendation “it is crucial that health professionals establish awareness campaigns focused on men’s family planning options, to educate men about vasectomy and other family planning options. This would begin to address the need to involve men in reproductive health and improve the acceptability and uptake of vasectomy. Furthermore, health promotion strategies should respond to cultural beliefs and societal norms in an undertaking to involve men in FP issues.” However; the recommendation is not helpful and simply very general. Please detail what kind of strategy? What are your other suggestions related to your findings for this specific society? Please highlight the key messages in the conclusion session. It would be very nice if authors elaborate implications to practice, research, or professional development. I also think that a section on implications should be added and concluding comments strengthened that would link this study to the health care professionals. It would benefit from expanding a little on how these findings could be used to operationalize improvements to men's health care in the international area.

We have tried our outmost best to respond to all the comments in the paragraph; we added text to elaborate of the recommendation as well as subheading. We also added text on lessons learned and implications for service rendering.

Reviewer 2 Report

The study is relevant, mainly in a place with low rates of vasectomy. However, some parts of the manuscript are not clear enough and create confusion. Some suggestions are:

I am confused about the objective of the study because of:

- It says that the aim was to explore the views about the barriers against the acceptability of vasectomy… Did the authors explore the views about “the barriers…” or the views about vasectomy?

- After reading the manuscript, I do not consider that the “knowledge and awareness of the men about vasectomy” has been evaluated.

- In the methodology section, the authors said that “The study also aimed to examine the barriers against and facilitators for the uptake of vasectomy”. Once again, I do not consider that the goal has been achieved

Methodology:

- I wonder if the clinics in which the data collection was carried out were private or public. Which is the socioeconomic level of men who attend these clinics? Which are the sociodemographic differences of the population that attend the rural, semi-rural, or urban clinic?

- The fact that the moderator of the focus groups was a woman, could affect the openness of the participants to discuss the issue of vasectomy?

Results:

- Were questions about the contraceptive methods used by the participants' partners? If not, I suggest commenting it on the limitations of the study.

- It would be interesting to separate the results according to the educational level of the participants, and according to their location (rural, semirural urban area).

- The thematic analysis yielded 5 themes and 14 subthems. However, the results are divided in 6 themes (not 5), and I couldn’t see the subthems.

Discussion:

- I think that the authors could discuss more about the implications of the study in the sociocultural context of Eswatini.

Author Response

Response to reviewer’s comments

We thank the reviewers for their valuable comments, we have responded to all the comments highlighted in this document and in the document. All revisions or additions are highlighted in blue.

 Reviewer 2

The study is relevant, mainly in a place with low rates of vasectomy. However, some parts of the manuscript are not clear enough and create confusion. Some suggestions are:

I am confused about the objective of the study because-It says that the aim was to explore the views about the barriers against the acceptability of vasectomy… Did the authors explore the views about “the barriers…” or the views about vasectomy?  In the methodology section, the authors said that “The study also aimed to examine the barriers against and facilitators for the uptake of vasectomy”. Once again, I do not consider that the goal has been achieved. After reading the manuscript, I do not consider that the “knowledge and awareness of the men about vasectomy” has been evaluated.

Thanks for the observation; while the problem is poor uptake of vasectomy in the country, the aim of the study was to explore the views of Eswatini men about the acceptability of vasectomy as a family planning option. We found that the participants lacked knowledge and awareness about vasectomy. We understand the views of the reviewer and have revised the aim of the study.

Methodology:

I wonder if the clinics in which the data collection was carried out were private or public. Which is the socioeconomic level of men who attend these clinics? Which are the sociodemographic differences of the population that attend the rural, semi-rural, or urban clinic?

We added text to clarify that of the three study sites, the first was a public facility located in a rural setting, the second was public clinic in semi-urban area, and thee third was a privately owned men’s urban clinic

The fact that the moderator of the focus groups was a woman, could affect the openness of the participants to discuss the issue of vasectomy?

We provided details on the procedure followed to recruit the men for the focus group discussion. Flyers with health promotion messages on FP methods, on vasectomy as a FP option, as well as a chart on vasectomy surgical procedure were distributed in the study sites before the schedule date for data collection. On the days of data collection, the first author delivered the talks and distributed brochures on vasectomy in English and SiSwati languages. After the talks, she was the one who recruited the participants who volunteered.

Results:

Were questions about the contraceptive methods used by the participants' partners? If not, I suggest commenting it on the limitations of the study.

We asked the participants about their views about family planning in general and about vasectomy. We added a theme to reflect their views about family planning.

It would be interesting to separate the results according to the educational level of the participants, and according to their location (rural, semirural urban area).

We did not plan to analyze data according to the geographic settings but we added the setting on the descriptors in the data presentation. For example, (FGD 8 urban men’s clinic, 48 years old) for all the quotations

The thematic analysis yielded 5 themes and 14 sub themes. However, the results are divided in 6 themes (not 5), and I could not see the sub themes.

We apologise for the oversight, we have corrected this and show that there were eight themes after revising the themes in line with the comments from both reviewers.

Discussion:

I think that the authors could discuss more about the implications of the study in the sociocultural context of Eswatini

We have revised the discussion and limited the focus on culture based on the comments from both reviewers.

Round 2

Reviewer 1 Report

Thank you for revising the manuscript. Most of my concerns were clarified. However, there are some serious reservations related to the methodology of the study. I summarise my review below for your information:

In the introduction and the discussion section, information about countries reproductive statistics such as “What is the incidence rate of contraceptive use among women and men?”, “the prevalence of other men related contraceptive methods such as condom and withdrawal”, “fertility rate in the country”, “What is the health policy about vasectomy in the country” etc, are still missing. In the methodology section, authors stated that sample were chosen from the clinics for the infertile patients or patients with erectile dysfunction problems “The clinic is reserved exclusively for male clients and provides services such as voluntary medical male circumcision, and infertility and erectile dysfunction services”. If the purpose of the study is to assess acceptability of vasectomy as a FP option, this centers were not appropriate to recruit participants. How come we can asses a mans opinion about vasectomy if he is infertile or having problems with erection?.   Needs to be clarified. Page 3 , line 121 “Flyers with health promotion messages on FP methods, on vasectomy as a FP option, as well as a chart on vasectomy surgical procedure were distributed in the study sites before the schedule date for data collection. These flyers were used in morning health talks that are routinely delivered in the morning to clients. On the days of data collection, the first author delivered the talks. After the talk, brochures on vasectomy were distributed to those that were literate; the brochures were in English and SiSwati languages”. It is not clear. What is the rationale for this? To inform participants before the data collection ? Page 8 line 343 The authors mentioned that in one of the participants “The health sector should deploy male nurses to assist male clients so that they are free to talk about male problems, not the very young nurses deployed to the clinics making it hard for male clients”-----The manuscript needs some information about the structure of the clinics such as number of health professionals and their gender in the methodology section.

Best.

Author Response

Response to reviewer

We thank the reviewer for their comments. We have highlighted the corrections in the document below and in the manuscript.

In the introduction and the discussion section, information about countries reproductive statistics such as “What is the incidence rate of contraceptive use among women and men?”, “the prevalence of other men related contraceptive methods such as condom and withdrawal”, “fertility rate in the country”,

We apologize for the oversight, we indicated that in 2018, the overall contraceptive use for any method of all women of reproductive age in Eswatini was 51.3% and 65% for married women with fertility rate of 3.3 births per woman [22]. The FP methods that a widely used are the injectable (27.6%), oral pills (13%), male condoms (48.1%), implant (5.1%), as well as the bilateral tubal ligation (4.7%) (Line 85-89)

“What is the health policy about vasectomy in the country” etc., are still missing.

In our response to the initial reviewer comments, we indicated that there is no policy for vasectomy in the country. We stated that the country has a National SRH Policy but the policy does not refer to the involvement of men in SRH and there is no mention of vasectomy et al (Line 80-82).

In the methodology section, authors stated that sample were chosen from the clinics for the infertile patients or patients with erectile dysfunction problems, “The clinic is reserved exclusively for male clients and provides services such as voluntary medical male circumcision, and infertility and erectile dysfunction services”. If the purpose of the study is to assess acceptability of vasectomy as a FP option, these centers were not appropriate to recruit participants. How come we can assess a man’s opinion about vasectomy if he is infertile or having problems with erection? Needs to be clarified.

In our initial response we stated that the study setting included three health facilities, the first data collection site was a public facility located in a rural setting in Hhohho region. The second site was a satellite public clinic in a densely populated semi-urban area, also in Hhohho region. The third data collection site was a privately owned urban clinic located in Manzini region.We apologize for the confusion, we stated that the third study-setting was a men’s only clinic the others are public facilities. Over all, the men's clinic provides medical male circumcision services, TB screening and treatment, STIs, family planning for males, curative, and laboratory services (Line 121-123).

Page 3 , line 121 “Flyers with health promotion messages on FP methods, on vasectomy as a FP option, as well as a chart on vasectomy surgical procedure were distributed in the study sites before the schedule date for data collection. These flyers were used in morning health talks that are routinely delivered in the morning to clients. On the days of data collection, the first author delivered the talks. After the talk, brochures on vasectomy were distributed to those that were literate; the brochures were in English and SiSwati languages”. It is not clear. What is the rationale for this? To inform participants before the data collection?

The health talks were used as a means to create awareness about vasectomy since there was no indication from the health facilities of health promotion activities on vasectomy. This step was necessary to ensure that the participants volunteer to participate in interviews about something they were aware of. As mentioned, Eswatini, the cultural practices and norms eschew open discussion of sexual matters, it was therefore important for the participants to volunteer to participate in the study after being well informed.

Page 8 line 343, the authors mentioned that in one of the participants “The health sector should deploy male nurses to assist male clients so that they are free to talk about male problems, not the very young nurses deployed to the clinics making it hard for male clients”. The manuscript needs some information about the structure of the clinics such as number of health professionals and their gender in the methodology section.

We would like to mention that it is very difficult to find information in Eswatini. Nevertheless, we indicated that the MoH states that significantly larger proportion of health workers in urban and rural facilities are female, this is of particular significance among nurses and midwives, a female predominant profession.  There is no documentation of staffing norms, number, gender, and professional category of staff working in health care facilities. However, the men’s clinic is manned by four nurses; two males and two female nurses, one male doctor and male phlebotomists (Page 126-130).

Round 3

Reviewer 1 Report

There is a contradictory between a statement about the prevalance rate of Vasectomy uptake in abstract (page 1 line 11) and in the introduction section (page 2 line 40). Please recheck and correct  

Please recheck your references,  There are some errors when citing articles. There is no information related to countries vasectomy uptake rates in two references ( 2,7) cited on page 2 line 39-40. 

The reference for United Nations (reference 4 in the reference list) is an institution, it is not an author as mentioned in the reference list as "Nations, U". needs to be corrected. 

Please delete the sentence on page 3 line 121 (dublication)

The statement on page 10 line 420-421  "Likewise, religious beliefs were the main  barrier against decision making in FP even in developed countries" is very pretentious since the cited reference was done in Malaysia which is not a developed country. Please consider rewording the statement or deleting it. 

The sentence on page 10 line 432-432 is not Clear. please consider revising the sentence. The study is subject to some limitations; one of the key limitation is that although it was limited 431 to adult men of 35 years and older including single or without partner.

Please consider deleting the sentence on page 10 line 432-437 starting with "Though the inclusion of these groups of men was purposive, the desire to have children have children might have skewed the acceptability of vasectomy among the study sample. We noted that the poor acceptability of vasectomy was observed among younger men who desire to have children. 
The findings should inform the development of messaging for the adoption of vasectomy for different population groups. A one-size fit all approach will not be effective."

Best 

Author Response

All the comments has been addressed and highlighted in the relevant sections of the manuscript.

There is a contradictory between a statement about the prevalence rate of Vasectomy uptake in abstract (page 1 line 11) and in the introduction section (page 2 line 40). Please recheck and correct

Response-corrected, we used the prevalence of 0.03% throughout the document

Please recheck your references; there are some errors when citing articles. There is no information related to countries vasectomy uptake rates in two references (2, 7) cited on page 2 line 39-40.

Response; reference no 7 has been deleted

The reference for United Nations (reference 4 in the reference list) is an institution; it is not an author as mentioned in the reference list as "Nations, U". Needs to be corrected.

Response; corrected

Please delete the sentence on page 3 line 121 (duplication)

Response; we deleted the sentence

The statement on page 10 line 420-421 "Likewise, religious beliefs were the main barrier against decision making in FP even in developed countries" is very pretentious since the cited reference was done in Malaysia, which is not a developed country. Please consider rewording the statement or deleting it.

Response; we rephrased the sentence and added an additional reference

The sentence on page 10 line 432-432 is not clear. Please consider revising the sentence. The study is subject to some limitations; one of the key limitation is that although it was limited 431 to adult men of 35 years and older including single or without partner.

Response: we rephrased the sentence

Please consider deleting the sentence on page 10 line 432-437 starting with "Though the inclusion of these groups of men was purposive, the desire to have children have children might have skewed the acceptability of vasectomy among the study sample. We noted that the poor acceptability of vasectomy was observed among younger men who desire to have children. The findings should inform the development of messaging for the adoption of vasectomy for different population groups. A one-size fit all approach will not be effective."

Response; the sentence is deleted